# Discovery, Quantitative Recurrence, and Inhibition of Motion-Blur Hysteresis Phenomenon in Visual Tracking Displacement Detection

**DOI:** 10.3390/s23198024

**Published:** 2023-09-22

**Authors:** Lixiang Shi, Jianping Tan

**Affiliations:** School of Mechanical and Electrical Engineering, Central South University, Changsha 410006, China; jptan@csu.edu.cn

**Keywords:** motion blur, point spread function, motion-blur hysteresis phenomenon, visual tracking, displacement detection

## Abstract

Motion blur is common in video tracking and detection, and severe motion blur can lead to failure in tracking and detection. In this work, a motion-blur hysteresis phenomenon (MBHP) was discovered, which has an impact on tracking and detection accuracy as well as image annotation. In order to accurately quantify MBHP, this paper proposes a motion-blur dataset construction method based on a motion-blur operator (MBO) generation method and self-similar object images, and designs APSF, a MBO generation method. The optimized sub-pixel estimation method of the point spread function (SPEPSF) is used to demonstrate the accuracy and robustness of the APSF method, showing the maximum error (ME) of APSF to be smaller than others (reduced by 86%, when motion-blur length > 20, motion-blur angle = 0), and the mean square error (MSE) of APSF to be smaller than others (reduced by 65.67% when motion-blur angle = 0). A fast image matching method based on a fast correlation response coefficient (FAST-PCC) and improved KCF were used with the motion-blur dataset to quantify MBHP. The results show that MBHP exists significantly when the motion blur changes and the error caused by MBHP is close to half of the difference of the motion-blur length between two consecutive frames. A general flow chart of visual tracking displacement detection with error compensation for MBHP was designed, and three methods for calculating compensation values were proposed: compensation values based on inter-frame displacement estimation error, SPEPSF, and no-reference image quality assessment (NR-IQA) indicators. Additionally, the implementation experiments showed that this error can be reduced by more than 96%.

## 1. Introduction

Image-based detection plays an increasingly important role in today’s society. The motion blur caused by the relative motion between the imaging device and the object during the exposure time can cause a decrease in image quality and affect subsequent detection. In fact, the processing of image degradation caused by motion blur has always been a very important issue in the detection of various fields such as satellite remote sensing [1], medical treatment [2], industry [3], and transportation [4]. Addressing the impact of motion blur on visual tracking and displacement detection can increase the application scenarios, reduce the demand for shooting equipment, and improve detection accuracy and robustness. The current solution mainly has two directions: (1) improving image degradation algorithms, and (2) improving visual tracking algorithms.

Firstly, in the field of image degradation, the process of image degradation is generally described as the result of the convolution of a clear image and a fuzzy kernel under certain noise interference [5,6]:(1)$$g\left(x,y\right)=f\left(x,y\right)⨂k+n\left(x,y\right)$$
where k is the point spread function (PSF), n represents random noise, g represents the degraded image, f is a clear image, and ⨂ is a convolution operation. Given g, the operation of estimating f is called image restoration or image deblurring, which can be divided into two kinds of problems: image non-blind deblurring and image blind deblurring. The difference lies in whether the point spread function k is known.

In the blind deblurring problem of images, as both f and k are unknown, this is a highly pathological problem that can have infinite solutions. Most existing research methods utilize a series of statistical prior knowledge from comparing clear images with blurred images, as well as prior knowledge from fuzzy kernels, to constrain these solutions, and to design a regularized loss function to iteratively solve f and k. The statistical priori of clear images and fuzzy images include image gradient sparse priori [7], natural image heavy tail gradient distribution priori [8], image edge gradient statistical priori [9], dark channel sparse priori (DCP) [10], bright channel sparse priori (BCP) [11], multi-scale structure self-similarity priori [12], etc. This kind of algorithm can achieve the deblurring of blurred images with complex nonlinear PSF. However, the calculation time is generally long, so it is not suitable for real-time tracking application. Some researches apply channel analysis techniques to images, such as cepstrum-based PSF estimation [13] and gradient domain dependent PSF estimation [5]. The cepstrum-based PSF estimation algorithm is easy to implement and short in time, but there is interference caused by the central cross bright fringe due to the boundary cutoff. The gradient domain autocorrelation-based PSF estimation can effectively reduce this interference [5].

Secondly, visual tracking refers to a series of image processing of moving targets in the image sequence to obtain the moving parameters, so as to carry out higher-level detection tasks. The general flow chart of visual tracking is shown in Figure 1. According to its development history, visual tracking can be roughly divided into three periods. The first period is roughly before 2011, which is called the classical tracking algorithm period, mainly the generation methods, such as the particle filter [14], meanshift [15], Kalman filter [16], and optical flow method [17]. From 2011 to 2015, it has been called the period of the correlation filtering tracking algorithm, which starts from MOSSE [18], CSK [19], and KCF [20], applying correlation filtering to the observation model of visual tracking. A generator based on a cyclic matrix is proposed, which solves the problem of intensive sampling perfectly mathematically. Moreover, the Fourier transform characteristic of the cyclic matrix is utilized to increase the training samples, but greatly shortens the training time. This has been followed by many more practical tracking algorithms such as DSST [21], SAMF [22], Staple [23], and LCT [24]. From 2015 to now, it has been called the tracking algorithm period of coexistence and fusion of correlation filtering and deep learning. Thanks to the ResNet [25] residual structure, the network depth is greatly improved, and the effect of the neural network in the feature extractor is greatly improved. Combined with the improvement of the CNN method and the development of high-performance computing hardware, deep learning is becoming increasingly influential in visual tracking, mainly represented by the ECO [26]-ATOM [27]-DiMP [28] series, Siamese [29,30,31,32] series, SSD series [33], YOLO series [34,35,36,37], and Vision transformer series [38,39], and is significantly improving the accuracy and robustness of tracking and detection results.

However, in this research, such a phenomenon, which also has an impact on the accuracy of visual tracking displacement detection, was found. When the visual tracking method is used to detect the displacement of a moving object, if the object starts to accelerate from standstill, the velocity data detected by the image-based tracking method are always slightly slower than that of the sensor used as the calibration, but when the object reaches the maximum velocity, the two pieces of velocity data are consistent, no matter which visual tracking method is used. At first, it was regarded as the image tracking detection error, but after analyzing the detection video frame by frame, it was found that this error should not be generated stably. We name this hysteresis phenomenon in the visual tracking of objects with motion blur during acceleration as the motion-blur hysteresis phenomenon (MBHP).

At present, there is almost no literature to explain and quantify this phenomenon. In addition, MBHP also has an impact on the annotation of image datasets, making the annotation of the existing datasets differ from the actual location of the target within them. In order to restore and quantify MBHP and increase the accuracy of image tracking and detection, this work explored MBHP in three steps: (1) re-modeling the video shooting process, optimizing the generation method of the motion-blur operator, and making a motion-blur image dataset which has images with strong motion blur while its annotation is very accurate, so it can be used to restore and quantify MBHP (in Section 3); (2) quantifying the impact of MBHP using different vision tracking methods (in Section 4); (3) designing methods to suppress this phenomenon from different perspectives (in Section 5). The main contributions from this work include the following:Obtaining a more accurate generation method of the motion-blur operator, making the simulation of motion blur more accurate;Proposing and constructing a more accurate motion-blur dataset;Discovering and quantifying the motion-blur hysteresis phenomenon in visual tracking displacement detection;Obtaining methods to suppress this phenomenon and improve video tracking accuracy and robustness.

## 2. Related Works

### 2.1. The Calculation of the PSF of Motion Blur

Assuming that image $f\left(x,y\right)$ moves in the plane, $v_{x}$ and $v_{y}$ are components of the moving velocity in the x and y directions, respectively. By ignoring noise interference and setting the exposure time as $T_{e}$, then the image after motion blur can be defined as:(2)$$g\left(x,y\right)=\int_{0}^{T_{e}}f(x-v_{x}t,y-v_{y}t)dt$$
where $g\left(x,y\right)$ is the image with motion blur. Fourier transforms both sides of Formula (2) and changes the order of integration:(3)$$G\left(u,v\right)=F(u,v)\int_{0}^{T_{e}}e^{-j2\pi (\frac{uv_{x}t}{M}+\frac{vv_{y}t}{N})}dt$$
where *M* and *N* are dimensions of the image $f\left(x,y\right)$, and let:(4)$$H\left(u,v\right)=\int_{0}^{T_{e}}e^{-j2\pi (\frac{uv_{x}t}{M}+\frac{vv_{y}t}{N})}dt$$

Then:(5)$$G\left(u,v\right)=F(u,v)H\left(u,v\right)$$

Formula (5) is the Fourier transform of Formula (1) ignoring noise. If the motion during the exposure time is simplified to uniform linear and the fractional displacement in the horizontal and vertical directions is $L_{x}$ and $L_{y}$, then Formula (4) can be calculated as:(6)$$H\left(u,v\right)=T_{e}e^{-j\pi \left(\frac{uL_{x}}{M}+\frac{vL_{y}}{N}\right)}\frac{\sin\left(\pi \left(\frac{uL_{x}}{M}+\frac{vL_{y}}{N}\right)\right)}{\pi \left(\frac{uL_{x}}{M}+\frac{vL_{y}}{N}\right)}\;=T_{e}e^{-j\pi \left(\frac{uL_{x}}{M}+\frac{vL_{y}}{N}\right)}sinc(\pi \left(\frac{uL_{x}}{M}+\frac{vL_{y}}{N}\right))$$
where $T_{e}e^{-j\pi \left(\frac{uL_{x}}{M}+\frac{vL_{y}}{N}\right)}$ is the constant term, $sinc(\pi \left(\frac{L_{x}}{M}u+\frac{L_{y}}{N}v\right))$ is an impulse signal, and the pulse impulse signal and the DC signal are mutually transformed through an inverse Fourier transform. Therefore, the PSF of motion blur can be approximately defined as:(7)$$h\left(x,y\right)=\left\{\begin{array}{c} \frac{1}{L},\;\;\sqrt{x^{2}+y^{2}}\leq L\;and\;\frac{x}{y}=-tan(\theta )\\ 0,\;\;\;\;\;others\;\;\;\;\;\;\;\;\;\;\;\;\;\;\;\;\;\;\;\; \end{array}\right.$$
where L is the motion-blur length, that is, the pixel distance at which an object moves on the image scale during the exposure time, and θ is the motion-blur angle, that is, the angle between the motion direction and the horizontal direction. Considering the two-dimensional discretization in the image, the PSF of motion blur is defined in Matlab as:(8)$$h\left(x,y\right)=Norm(max(1-d_{nearest},0))$$
where $d_{nearest}$ is the distance between the position $\left(x,y\right)$ and the ideal line segment determined by the motion-blur length L and the motion-blur angle θ, $\max\left(.\right)$ is the maximum function, and Norm(.) is a normalized function. Moreover, in order to facilitate the generation and restoration of a motion-blurred image, the center of the PSF is taken as the center of the ideal line segment, so the size of the PSF is optimized to be odd.

Taking a motion-blur length of 6 px and a motion-blur angle of 45° as an example, the motion-blur operator (MBO) generated by Formula (8) with L=6, θ=45 is shown in Table 1.

This PSF generation method is widely used due to its simple principle and ease to implement. However, it is approximate and can be used in situations with low accuracy requirements. Additionally, it is clearly not sufficient for accurately quantifying MBHP. To obtain a more accurate MBO, it is necessary to re-model the motion-blur process.

### 2.2. PSF Estimation Method Based on Cepstrum

Cepstrum was first used in earthquake and bomb echo detection, and it is now widely used in signal processing fields, such as speech analysis and conversion and biomedical image processing [13]. Cepstrum is convenient to extract and analyze the periodic signals which are difficult to be recognized by the naked eye on the original spectral diagram, and it can simplify the clustered edge band spectral lines on the original spectral diagram into a single spectral line, which is less affected by the sensor’s measuring point position and transmission path. The cepstrum of an image is actually short for logarithmic cepstrum, also known as the logarithmic power spectrum, which can be calculated as follows:(9)$$C_{f}\left(p,q\right)=F^{-1}(\mathrm{lg}\left(P_{f}\left(u,v\right)\right))$$
where $F^{-1}(.)$ is the inverse Fourier transform, lg(.) is a logarithmic function, and $P_{f}(u,v)$ is the power spectral density function of image $f\left(x,y\right)$. Since an image is a truncated finite signal, it can be regarded as an energy signal, so its power spectrum can be estimated as follows:(10)$$P_{f}(u,v)=\frac{{\left|F\left(f\left(x,y\right)\right)\right|}^{2}}{T}$$
where F(.) is the Fourier transform and T is the total number of pixels of image $f\left(x,y\right)$. Since the images being compared and analyzed are generally of the same size, it is worth noting that the relative value of the image cepstrum, constant coefficient, and constant term can be ignored. In addition, the zero frequency point is then moved to the middle of the spectrum to obtain the commonly used cepstrum formula of image $f\left(x,y\right)$:(11)$$C_{f}\left(p,q\right)=FShift(F^{-1}\left(\mathrm{lg}\left(\left|F\left(u,v\right)\right|+1\right)\right))$$
where FShift(.) represents moving the zero frequency point to the center of the spectrum. The logarithmic operation adds one to ignore the details of the low amplitude. The distance and angle between the minimum value in the cepstrum $C_{f}\left(p,q\right)$ and its image center are the motion-blur length and motion-blur angle of the PSF analyzed by the cepstrum. That is:(12)$$L_{e}=\sqrt{{\left(p_{mid}-p_{min}\right)}^{2}+{\left(q_{mid}-q_{min}\right)}^{2}}$$
(13)$$\theta_{e}=\left\{\begin{array}{c} \arctan\left(\frac{q_{mid}\;-\;q_{min}}{p_{mid}\;-\;p_{min}}\right),\;\;p_{mid}!=p_{min}\;\\ \;\;\;\;\;\;90{}^{\circ}\;\;\;\;\;\;\;\;,\;\;\;p_{mid}=p_{min} \end{array}\right.$$
where $p_{mid},\;q_{mid}$ are the transverse and vertical coordinates of the center of $C_{f}\left(p,q\right)$, $p_{smin}$, $q_{smin}$ are the transverse and vertical coordinates of the minimum value of $C_{f}\left(p,q\right)$, $L_{e}$ is the motion-blur length estimated by cepstrum analysis, $\theta_{e}$ is the motion-blur angle estimated by cepstrum analysis, and arctan(.) is the inverse tangent function.

It is worth noting that although this method can resist interference, it can only obtain pixel level accuracy detection results. To obtain sub-pixel accuracy detection results, it is necessary to optimize it.

### 2.3. No-Reference Image Quality Assessment Indicators

Image quality assessment (IQA) is generally defined as evaluating the degree of visual distortion of an image by analyzing the relevant characteristics of the image signals. It can be divided into a full-reference image quality assessment (FR-IQA), semi-reference image quality assessment (SR-IQA), and no-reference image quality assessment (NR-IQA) [40]. Compared to an FR-IQA, an NR-IQA does not require any reference information when calculating the visual quality of distorted images, and has a wider application prospect in practical application systems. It has been extensively studied and used in the autofocus technology of mobile phones and cameras [41]. An NR-IQA can be divided into two categories: (1) based on the spatial distribution angle and (2) based on the gradient. The difference between the motion-blur and clear images is that the gradient of the object in the direction of motion on the motion-blur image decreases, making it suitable for gradient-based NR-IQA indicators. The commonly used NR-IQA indicators [42] are as follows.

Brenner Gradient function:
(14)$$D_{B}\left(f\right)=\sum_{y}\sum_{x}{\left|f\left(x+2,y\right)-f(x,y)\right|}^{2}$$Tenengrad Gradient function:
(15)$$D_{T}\left(f\right)=\sum_{y}\sum_{x}\left|G\left(x,y\right)\right|\;\;\;\left(G\left(x,y\right)>Threshold\right)$$
(16)$$G\left(x,y\right)=\sqrt{G_{x}^{2}\left(x,y\right)+G_{y}^{2}(x,y)}$$
where Threshold is the given edge detection threshold, and $G_{x}$ and $G_{y}$ are the convolution of the Sobel horizontal and vertical edge detection operators at pixel point $\left(x,y\right)$, respectively. The proposed Sobel operator is as follows:(17)$$g_{x}=\frac{1}{4}\left[\begin{array}{ccc} -1 & 0 & 1\\ -2 & 0 & 2\\ -1 & 0 & 1 \end{array}\right],\;\;\;g_{y}=\frac{1}{4}\left[\begin{array}{ccc} 1 & 2 & 1\\ 0 & 0 & 0\\ -1 & -2 & -1 \end{array}\right]$$Laplacian Gradient function:
(18)$$D_{L}\left(f\right)=\sum_{y}\sum_{x}\left|G\left(x,y\right)\right|\;\;\;\left(G\left(x,y\right)>Threshold\right)$$
where $G\left(x,y\right)$ is the convolution of the Laplacian operator at pixel point $\left(x,y\right)$, and Threshold is the given edge detection threshold.Gray difference function SMD:
(19)$$D_{S}\left(f\right)=\sum_{y}\sum_{x}\left(\left|f\left(x,y\right)-f\left(x,y-1\right)\right|+\left|f\left(x,y\right)-f\left(x-1,y\right)\right|\right)$$Gray difference product function SMD2:
(20)$$D_{S2}\left(f\right)=\sum_{y}\sum_{x}\left(\left|f\left(x,y\right)-f\left(x,y-1\right)\right|\times \left|f\left(x,y\right)-f\left(x-1,y\right)\right|\right)$$Energy gradient function EOG:
(21)$$D_{E}\left(f\right)=\sum_{y}\sum_{x}\left({\left(f\left(x,y\right)-f\left(x,y-1\right)\right)}^{2}+{\left(f\left(x,y\right)-f\left(x-1,y\right)\right)}^{2}\right)$$

## 3. Methods and Dataset

### 3.1. The More Accurate MBO Generation Method: APSF

As mentioned in Section 2.1, Formula (8) is an approximate result of true MBO and its error will affect the observation results of MBHP. In order to obtain a more accurate two-dimensional discretization motion-blur operator so as to describe motion ambiguity more accurately, as shown in Figure 2, one-dimensional movement was first studied.

Assuming that the photographed object moves from top to bottom, a pixel box P of one frame (Frame i) is taken for observation. Figure 2a is obtained, the interval between two adjacent frames is $T_{S}$, and the shutter time of the camera is $T_{e}$. Considering relative motion, the bottom-up movement of the object is equivalent to the top-down movement of the observation frame. Considering that the position at which the shutter closes is mapped between the discrete image space-time and the real world space-time, the moment at which the shutter closes for each frame can be defined as the moment at which each frame ends. As shown in Figure 2b, the observed pixel box moves from position $P_{1}$ to position $P_{3}$ within $T_{S}.$ Obviously, the images from position $P_{2}$ to position $P_{3}$ will be superimposed together to generate pixel box P in Frame i + 1. If the image is discrete along the motion direction into an infinitesimal rectangle, its motion-blur operator will be the isosceles trapezoid, as shown in Figure 2c. If the image pixel position is discrete, the proportion of each pixel box will be the result of the normalization of the graph area shown in Figure 2d. If the exposed area of the discrete pixel frame in the exposure time is directly considered, it is the result of the normalization of the graph area shown in Figure 2e. Figure 2d,e are equivalent, and the result is summarized and defined as follows:(22)$${h\left(y\right)}_{L}=Norm(\left\{\begin{array}{c} \;\;1\;\;\;\;\;\;\;\;\;\;\;\;\;\;L_{m}=1\;or\;L_{m}\geq 3,\;y=1\;\;\\ 2d_{L}-{d_{L}}^{2}\;\;\;\;\;\;\;\;\;L_{m}=2,\;y=1\;\;\;\;\;\;\;\;\;\;\;\;\;\;\;\\ 2\;\;\;\;\;\;\;\;\;\;\;\;\;\;L_{m}\geq 3,\;1<y<L_{m}-1\;\;\\ 1+2d_{L}-{d_{L}}^{2}\;\;\;\;\;\;\;L_{m}\geq 3,\;y=L_{m}-1\;\;\;\;\;\;\;\;\;\;\;\\ {d_{L}}^{2}\;\;\;\;\;\;\;\;\;\;\;\;\;y=L_{m}\;\;\;\;\;\;\;\;\;\;\;\;\;\;\;\; \end{array})\right.$$
where $d_{L}$ is the fractional part of the motion-blur length L, and is not 0, defined in Formula (23), $L_{m}$ is the size of the motion-blur operator, defined in Formula (24), and Norm(.) is a normalized function.
(23)$$d_{L}=\left\{\begin{array}{c} L-floor\left(L\right)\;\;\;\;\;\;\;\;\;\;\;\\ \;1\;\;if\;\;L-floor\left(L\right)==0 \end{array}\right.$$
(24)$$L_{m}=ceil\left(L\right)+1$$
where floor(.) is an integer down function and ceil(.) is an integer up function.

For two-dimensional movement, there is such a prior: The result of summing the motion-blur operator in rows or columns for two-dimensional motion is equivalent to a one-dimensional motion blur operator constructed with the same motion-blur length in that direction, namely:(25)$${h_{x}\left(y\right)}_{L_{my}}=\sum_{x=1}^{L_{mx}}h\left(x,y\right)$$
(26)$${h_{y}\left(x\right)}_{L_{mx}}=\sum_{y=1}^{L_{my}}h(x,y)$$
(27)$$L_{mx}=ceil\left(L\times cos(\theta )\right)+1$$
(28)$$L_{my}=ceil\left(L\times sin(\theta )\right)+1$$

Then, each column of the motion-blur operator is analyzed, which is similar to Figure 2e of the decomposition process of one-dimensional motion-blur operator. The sum of the column is equal to the volume of a sloping triangular prism (isosceles triangle in contrast Figure 2e). As shown in Figure 3, the component of the row is equal to the ratio of the volume of the sloping triangular prism in the row to its total volume. It is worth noting that similar to the motion-blur operator of one-dimensional movement, the oblique triangular prisms (triangle in one-dimensional movement, Figure 2e) in the head position and end position are partial, Figure 3a corresponds to Figure 2e Q_1_, Figure 3b corresponds to Figure 2e Q_2_ and Q_3_, and Figure 3c corresponds to Figure 2e Q_4_. So, the motion-blur operator of two-dimensional movement should be calculated as follows:(29)$$h\left(x,y\right)=\frac{V_{x}\left(y\right)-V_{x}\left(y-1\right)}{V_{x}}{h_{y}\left(x\right)}_{L_{mx}}=(V_{xrate}\left(y\right)-V_{xrate}(y-1)){h_{y}\left(x\right)}_{L_{mx}}$$
where $V_{x}$ is the total volume of this column, $V_{x}\left(y\right)$ is the volume between rows 0 and y in the total volume of the column, and $V_{xrate}\left(y\right)$ is the volume ratio between rows 0 and y in the total volume of the column. Due to different slopes, starting and ending positions need to be considered. The principle of volume calculation is simple, but the specific situations that need to be discussed are relatively complex. Therefore, only an overview of the algorithm will be explained here. Firstly, given the motion-blur angle *θ* and the distance $c_{y}$ from the end position of the block passing through this row to row y, it is easy to classify and discuss the total volume $V_{x,[0,1]}$ and $V_{xrate}(\theta ,c_{y},[0,1])$ (Figure 3b). Similarly, it is easy to classify and discuss the total volume $V_{x,[0,0.5]}$ and $V_{xrate}(\theta ,c_{y},[0,0.5])$. Then, based on this, calculate the total volume $V_{x,[0,c_{x2}]}$ and $V_{xrate}(\theta ,c_{y},[0,c_{x2}])$ (Figure 3c), and then calculate the total volume $V_{x,[0.5,c_{x2}]}$ and $V_{xrate}(\theta ,c_{y},[0.5,c_{x2}])$ (Figure 3a). Here, $V_{x,\left[c_{x1},c_{x2}\right]}$ means the total volume of column x, which is also the maximum value of $V_{x,\left[c_{x1},c_{x2}\right]}(y)$, and $V_{x,\left[c_{x1},c_{x2}\right]}(y)$ is the expansion of $V_{x}\left(y\right)$. $V_{xrate}(\theta ,c_{y},[c_{x1},c_{x2}])$ is the expansion of $V_{xrate}\left(y\right)$. The values of $c_{x1}$ and $c_{x2}$ are the relative positions of the real starting and end positions of the block passing through the column being calculated.
(30)$$V_{x,[0.5,c_{x2}]}=V_{x,[0,c_{x2}]}-V_{x,[0,0.5]}$$
(31)$$V_{xrate}\left(\theta ,c_{y},\left[0.5,c_{x2}\right]\right)=\frac{V_{xrate}\left(\theta ,c_{y},\left[0,c_{x2}\right]\right)V_{x,\left[0,c_{x2}\right]}-V_{xrate}\left(\theta ,c_{y},\left[0,0.5\right]\right)V_{x,\left[0,0.5\right]}}{V_{x,[0.5,c_{x2}]}}$$

Taking a motion-blur length of 6 px and a motion-blur angle of 45° as an example, the MBO generated by APSF according to Formula (29) is shown in Table 2.

### 3.2. PSF Sub-Pixel Estimation Method: SPEPSF

The cepstrum estimation resolution of the PSF calculated by Formula (12) is at pixel level. For sub-pixel estimation, an alternative method is to extract the minimum channels in the two dimensions of the cepstrum image, similar to DCP method, and do fittings, respectively, to estimate the location of the minimum value. That is:(32)$${C_{f1}^{min}(p)=min1(C}_{f}\left(p,q\right))$$
(33)$${C_{f2}^{min}(q)=min2(C}_{f}\left(p,q\right))$$
(34)$${p_{smin}=min(FIT(C}_{f1}^{min}\left(p\right)))$$
(35)$${q_{smin}=min(FIT(C}_{f2}^{min}\left(q\right)))$$
where $C_{f}\left(p,q\right)$ is calculated by Formula (11), $min1\left(.\right)$ returns the column vector of the minimum value for each row, $min2\left(.\right)$ returns the row vector of the minimum value for each column, and FIT(.) is a fitting function. Then, the PSF sub-pixel estimation method (SPEPSF) is defined as:(36)$$L_{e}=\sqrt{{\left(p_{mid}-p_{smin}\right)}^{2}+{\left(q_{mid}-q_{smin}\right)}^{2}}$$
(37)$$\theta_{e}=\left\{\begin{array}{c} \arctan\left(\frac{q_{mid}\;-\;q_{smin}}{p_{mid}\;-\;p_{smin}}\right),\;\;p_{mid}!=p_{smin}\;\\ \;\;\;\;\;\;\;90{}^{\circ}\;\;\;\;\;\;\;\;,\;\;\;p_{mid}=p_{smin} \end{array}\right.$$
where $L_{e}$ is the sub-pixel motion-blur length estimated by SPEPSF and $\theta_{e}$ is the motion-blur angle estimated by SPEPSF.

### 3.3. A Fast Image Matching Method Based on Fast Correlation Response Coefficient: FAST-PCC

Correlation-based data matching is a commonly used method in information detection. MOSSE [18] introduces correlation filtering into the field of image matching detection. When there is light imbalance in the image, correlation-based matching is more effective than MSE-based matching. The correlation response used in general image matching uses covariance because when the image is large, calculating the correlation coefficient takes too long. Here, a method is used that can quickly calculate the correlation response coefficient. Firstly, the correlation coefficient used in image matching refers to the Pearson Correlation Coefficient (PCC), which is calculated as follows:(38)$$PCC\left(X,Y\right)=\frac{cov\left(X,Y\right)}{\sigma_{X}\sigma_{Y}}=\frac{\sum_{i=1}^{n}\left(X_{i}-E\left(X\right)\right)\left(Y_{i}-E\left(Y\right)\right)}{\sqrt{\sum_{i=1}^{n}{\left(X_{i}-E\left(X\right)\right)}^{2}}\sqrt{\sum_{i=1}^{n}{\left(Y_{i}-E\left(Y\right)\right)}^{2}}}\;=\frac{E\left(XY\right)-E(X)E(Y)}{\sqrt{E\left(X^{2}\right)-E^{2}(X)}\sqrt{E\left(Y^{2}\right)-E^{2}(Y)}}$$
where X and Y are one-dimensional vectors of n elements, $\sigma_{X}$ and $\sigma_{Y}$ are the standard deviation of X and Y, and cov(.) is the covariance function. In image matching, the correlation coefficient matrix of a template image matrix T and image matrix S needs to be calculated. Here, S is a two-dimensional or three-dimensional image, T is an image with a two-dimensional dimension not greater than S, and the third dimension is the same as S. When the dimensions of T and S are large, direct calculation can be time-consuming, and the accelerated calculation formula is as follows:(39)$$PCC(T,S)=\frac{convn\left(S,\frac{T_{r}}{N_{T}}\right)-E\left(T\right)convn(S,k_{T})}{\sqrt{\left(E\left({T.}^{2}\right)-E^{2}\left(T\right)\right)(convn\left({S.}^{2},k_{T}\right)-{\left(convn\left(S,k_{T}\right)\right).}^{2})}}$$
where
(40)$$E\left(T\right)=\frac{1}{N_{T}}\sum_{i=1}^{M}\sum_{j=1}^{N}\sum_{k=1}^{Q}T(i,j,k)$$
(41)$$E\left({T.}^{2}\right)=\frac{1}{N_{T}}\sum_{i=1}^{M}\sum_{j=1}^{N}\sum_{k=1}^{Q}T^{2}(i,j,k)$$
(42)$$k_{T}=ones(M,N,Q)/N_{T}$$
(43)$$T_{r}=imrotate(T,180{}^{\circ})$$
(44)$$N_{T}=MNQ$$
where M, N, and Q are, respectively, the 3D size of the template picture T. convn(.) is a multidimensional convolution function, and here it returns a matrix as large as the first parameter because the frequency domain product is used to calculate the time domain convolution, so there is a computational acceleration effect. ones(.) is a function that generates an all-1 matrix, imrotate(.) is the rotation of the image in one or two dimensions, and ${(.).}^{2}$ is an element-by-element squared operation.

If the addition, subtraction, multiplication, division, and square root operations are all recorded as one operation, the original algorithm would operate approximately $5N_{T}N_{S}+10N_{S}$ times. The Time complexity of the original algorithm is $O(N_{T}N_{S})$, while the accelerated algorithm uses the frequency domain to calculate convolution, which is about $10N_{S}+3N_{T}+7$ times, and the Time complexity is $O(N_{T}+N_{S})$, which greatly reduces the calculation time consumption.

### 3.4. Improved KCF

This work requires representative tracking methods to show the universality of MBHP. KCF [20] is such a method, but the original version of KCF is used for tracking and detecting general objects. It is not suitable for scenarios where high-speed motion, objects quickly leave the field of view and precise detection is required. There are several issues:KCF uses hog features and has better tracking performance than using original image features, but using fhog with a cell of 4 for calculation results in N-fold down-sampling for large-sized images and targets results in a resolution of 4n(px);The target quickly leaves the field of view and cannot be tracked and detected;Not suitable for tracking high-speed objects due to low speed and small range movement.

Therefore, several improvements have been made:Perform two-dimensional polynomial fitting on the detection response of KCF for the HOG feature to make it continuous;Change the tracking strategy for general objects and use a template strategy with a lifecycle of 1 frame, that is, update the template every frame;Strengthen the use of motion trackers.

### 3.5. Construction of Motion Blur Video Dataset

In order to restore MBHP, a motion-blur video dataset with accurate annotation is needed. However, MBHP will cause interference to the annotation of motion-blur video datasets, so the existing datasets, whether marked by human or machine, does not meet the requirement. Since the more accurate generation method of MBO was obtained in Section 3.1, a self-similar object photographed in reality was used to obtain a cycle graph by merging it from end to end after image processing. Then motion simulation with motion blur is carried out to generate the video dataset. The generation procedure of motion-blur video dataset generated from a self-similar object graph is shown in Figure 4. As the wire rope is wound by strands of steel wire or wire bundle according to certain rules, it is a typical self-similar object. Therefore, a real picture of steel wire rope is used as the representative of self-similar object to generate the dataset.

Figure 4 describes a general method of generating motion-blur video datasets through a self-similar image. Operations 1 and 9 in Figure 4 represent the input of the image and the output of the generated motion-blur video dataset.

Operation 2 is that the background of the image generally taken may be messy, which may interfere with the subsequent process, so it is necessary to separate the object from the background and add a clean background to the object. If the background of the image is clean, this step can be omitted, and the separation operation can be completed by using the commonly used image matting software such as Adobe Photoshop 2023. The image obtained in this step is $S_{2}\left(x,y\right)$.

Operation 3 is to rotate the object in the image $S_{2}\left(x,y\right)$ to vertical, which is a prior of utilizing the self-similar object characteristics of the wire rope: if the wire rope is vertical, then take an appropriate template on the wire rope image, and multiple peaks of the response coefficient related to the wire rope image will also be close to a vertical line. The resulting image is $S_{3}\left(x,y\right)$.

Operation 4 is to use projective mapping four-point method [43] to solve the perspective distortion problem caused by the camera not strictly perpendicular to the image plane. After obtaining the projective transformation matrix, the projective transformation is applied to $S_{3}\left(x,y\right)$ and interpolation is performed, the obtained image is clipped according to the requirements, copied, and stitched together to get the image $S_{4}\left(x,y\right)$.

Operation 5 is to carry out linear gradient transition operation at the stitching place to get the image $S_{5}\left(x,y\right)$. The effect is shown in Figure 5.

Operation 6 is to crop half of the image $S_{5}\left(x,y\right)$ to get $S_{6}\left(x,y\right)$, where the stitching part with linear gradient transition is in $S_{6}\left(x,y\right)$. $S_{6}\left(x,y\right)$ is a cyclable graph and is repeatedly self-spliced along the column direction to get $S_{6S}\left(x,y\right)$.

Operation 7 divides $S_{6}\left(x,y\right)$ into two parts: wire rope $S_{7r}\left(x,y\right)$ and background $S_{7b}\left(x,y\right)$, and likewise divides $S_{6S}\left(x,y\right)$ into two parts: wire rope $S_{7Sr}\left(x,y\right)$ and background $S_{7Sb}\left(x,y\right)$.

Operation 8 is to calculate the displacement according to the elapsed time, set frame rate and exposure time, and capture the image $S_{8r}\left(x,y\right)$ from $S_{7Sr}\left(x,y\right)$ according to the displacement. Transform $S_{8r}\left(x,y\right)$, respectively, according to no motion blur applied, light motion blur applied (short exposure time), and heavy motion blur applied (long exposure time). Overlay the results with background $S_{7b}\left(x,y\right)$ to obtain corresponding simulated images $S_{8row}\left(x,y\right)$, $S_{8blur1}\left(x,y\right)$, $S_{8blur2}\left(x,y\right)$ and $S_{8blur3}\left(x,y\right)$.

Steel wire ropes are applied in multiple fields, and in the lifting of ultra-deep mines, the lifting speed needs to reach 18 m/s [44]. The maximum lifting speed of the ultra-deep mine simulation lifting test bench is simulated at 10:1 is 1.8 m/s. Based on this, the motion trajectory is formulated, with a video capture frame rate of 30 Hz. Assuming that the maximum speed is achieved by uniformly accelerating from 0 for 5 s, and then running at a constant speed for 5 s, the speed is uniformly reduced to 0 within 5 s. The speed formula is as portrayed in Formula (45), and the speed map is shown in Figure 6a. The resolution of the image is 20 px/mm. The speed calculation for each frame is shown in Formula (46), and the speed map is shown in Figure 6b. Four exposure times are set for ideal situations of close to 0 s, 1/300 s, 1/100 s, and 1/60 s, respectively, corresponding to no motion blur, light motion blur, medium motion blur, and heavy motion blur. The length formulas for the last three types of motion blur are shown in Formula (47)–(49).
(45)$$v_{t}=\left\{\begin{array}{c} 0.36t\;\;\;\;\;\;\;\;0\leq t\leq 5\\ 1.8\;\;\;\;\;\;\;\;\;5<t\leq 10\\ 5.4-0.36t\;\;\;10<t\leq 15\; \end{array}\right.$$
(46)$$v_{f}=\left\{\begin{array}{c} \;8(n_{f}-1)\;\;\;\;\;\;\;1\leq n_{f}\leq 151\\ \;\;1200\;\;\;\;\;\;\;\;\;151<n_{f}\leq 301\\ 3608-8n_{f}\;\;\;301<n_{f}\leq 451\; \end{array}\right.$$
(47)$$L_{mb1}=0.1v_{f}$$
(48)$$L_{mb2}=0.3v_{f}$$
(49)$$L_{mb3}=0.5v_{f}$$

Integral of displacement per frame from initial position to $v_{f}$:(50)$$s_{f}(i)=\sum_{j=1}^{i}v_{f}(j)$$

The motion-blur operator of Frame i corresponding to the exposure duration of 1/300 s, 1/100 s, and 1/60 s is as follows:(51)$$h_{mb1,i}=APSF(L_{mb1}\left(i\right),-90)$$
(52)$$h_{mb2,i}=APSF(L_{mb2}\left(i\right),-90)$$
(53)$$h_{mb3,i}=APSF(L_{mb3}\left(i\right),-90)$$
where APSF(.) is the APSF method proposed in Section 3.1. The images of Frame i corresponding to the four exposure times are as follows:(54)$$S_{8row,i}=S_{7S}\left(s_{f}\left(i\right)+1:s_{f}\left(i\right)+M_{S},:,:\right)+S_{7b}\left(x,y\right)$$
(55)$$S_{8blur1,i}=convn(S_{7S}\left(s_{f}\left(i\right)+1-ceil(L_{mb1}\left(i\right)):s_{f}\left(i\right)+M_{S},:,:\right),h_{mb1,i})+S_{7b}\left(x,y\right)$$
(56)$$S_{8blur2,i}=convn(S_{7S}\left(s_{f}\left(i\right)+1-ceil(L_{mb2}\left(i\right)):s_{f}\left(i\right)+M_{S},:,:\right),h_{mb2,i})+S_{7b}\left(x,y\right)$$
(57)$$S_{8blur3,i}=convn(S_{7S}\left(s_{f}\left(i\right)+1-ceil(L_{mb3}\left(i\right)):s_{f}\left(i\right)+M_{S},:,:\right),h_{mb3,i})+S_{7b}\left(x,y\right)$$

A total of four groups of images can be obtained, which together form the motion-blur video dataset based on self-similar object images. It can be found that the motion of objects in these four groups of images is synchronous, only the motion-blur parameters are different, and the designed motion parameters are the measured values of the ideal displacement sensor in the real environment.

It is worth pointing out that the biggest difference between this motion-blur dataset and other datasets is that it has images with strong motion blur while its annotation is very accurate. So it can be used to restore and quantify MBHP.

## 4. Results

### 4.1. Error Comparison of PSF Generation Methods Based on SPEPSF

It is difficult to directly compare APSF and existing MBO generation methods (requiring one-to-one correspondence between motion-blur lengths and real motion-blur images, such as using laser displacement sensors to record object displacement while shooting moving objects, but unfortunately, this will introduce other errors). Since the PSF sub-pixel estimation method (SPEPSF) was optimized in Section 3.2 and, theoretically, SPEPSF can ensure the accuracy of motion operator estimations for images with simple motion blur, the SPEPSF method can be used to verify the effect of the proposed APSF method. Four methods of generating a two-dimensional PSF were compared in total. Method 1 is a two-dimensional PSF calculated according to Formula (8), Method 2 is the rotation of a one-dimensional PSF calculated according to Formula (8), Method 3 is the rotation of a one-dimensional PSF calculated by APSF, and Method 4 is a two-dimensional PSF calculated according to APSF. Since the motion-blur angle (90°-θ) and θ have similar effects, three representative angles, 0°, 30°, and 45°, between 0° and 45°, were taken as the simulated motion-blur angles. The length of motion blur increased from 1 px to 60 px with a resolution of 0.1 px. Four methods are used to calculate the two-dimensional PSF, and the PSF is estimated by the SPEPSF method. Error comparison indicators are maximum error (ME) and mean square error (MSE), defined in Formula (58)–(61). Additionally, the results are shown in Figure 7, Figure 8, Figure 9 and Figure 10 and Table 3.
(58)$${error}_{Li}=L_{ei}-L_{i},\;\;{ME}_{L}=max({error}_{Li})$$
(59)$${MSE}_{L}=\frac{1}{n}\sum_{i=1}^{n}{{error}_{Li}}^{2}$$
(60)$${error}_{\theta i}=\theta_{ei}-\theta_{i},\;\;\;{ME}_{\theta }=max({error}_{\theta i})$$
(61)$${MSE}_{\theta }=\frac{1}{n}\sum_{i=1}^{n}{{error}_{\theta i}}^{2}$$

From Figure 7 and Table 3, it can be observed that when the motion-blur angle is 0°, the estimated motion-blur length and angle curve of the PSF generated by Methods 1 and 2 after cepstrum analysis coincide. The estimated motion-blur length and angle curve of the PSF generated by Methods 3 and 4 after cepstrum analysis coincide. The MSE of Methods 3 and 4 is 65.67% lower than that of Methods 1 and 2, and as the motion-blur length increases, the effects of Methods 3 and 4 tend to be perfect, with a maximum error reduction of 86% from 0.29 px to 0.04 px for strong motion blur above 20 pixels.

From Figure 8 and Table 3 it can be observed that when the motion-blur angle is 30°, the PSF generated by Methods 1, 2, and 3 have significant errors in the estimated motion-blur length after cepstrum analysis, with maximum errors of 1.85 px, 1.45 px, and 1.94 px, respectively. However, the PSF generated by Method 4 has a maximum error of 0.84 px after cepstrum analysis, and the MSE of Method 4 is reduced by 49.47% compared to Method 1. Due to the large estimation error of the motion-blur length at small scales, the estimation error of the motion-blur angle at small scales is greater. When the motion-blur length is greater than 6 px, the estimation error of the motion-blur angle tends to be normal (as shown in Figure 10). At this time, the MSE of Method 4 is reduced by 42.25% compared to Method 1. For high-speed motion with strong motion blur greater than 20 pixels, the maximum error is reduced by 20% from 0.80 px to 0.64 px.

From Figure 9 and Table 3, it can be observed that when the motion-blur angle is 45°, the estimated motion-blur length error of the PSF generated by Methods 1 and 2 after cepstrum analysis has a significant fluctuation, followed by Method 3 with the smallest error fluctuation. The MSE of Method 4 decreases by nearly 77.82% compared to Method 1, and as the motion-blur length increases, the MSE of Method 4 tends to be at a minimum. For high-speed motion with strong motion blur greater than 20 pixels, the maximum error is reduced by 87% from 0.601 px to 0.079 px.

Therefore, from the perspective of testing the PSF generation methods based on cepstrum estimation, it is believed that the proposed Method 4 performs better than the widely used Method 1. It can be said that the PSF generated by the proposed Method 4 (APSF) tends to be more realistic.

### 4.2. The Recurrence and Quantification of Motion-Blur Hysteresis Phenomenon

#### 4.2.1. Motion-Blur Hysteresis Phenomenon in Visual Tracking Using FAST-PCC

The FAST-PCC method introduced in Section 3.3 is used to track each group of images in the motion-blur video dataset obtained in Section 3.5. Figure 11 shows the recorded error between the inter-frame velocity detected by this method and the set inter-frame velocity, as well as the overall displacement error.

By comparing the speed errors and displacement errors of different exposure times in Figure 11, it can be found that as the exposure time increases, there will be more significant speed errors and displacement errors during acceleration and deceleration. During the entire acceleration period, a maximum displacement error of −300 px was generated (when the exposure duration is 0.5 times that of the inter-frame time). During the uniform motion phase, no new displacement errors were generated, while during the deceleration phase, a maximum displacement error of 300 px was generated. After an acceleration uniform deceleration operation, the total error of the object tends to go to zero. The speed error and maximum displacement error during acceleration or deceleration are both related to the exposure time and have a nearly linear relationship. During acceleration, the speed error is negative, while during deceleration, the speed error is positive.

#### 4.2.2. Motion-Blur Hysteresis Phenomenon in Visual Tracking Using Improved KCF

The improved KCF method introduced in Section 3.4 using the HOG feature is used to track each group of images in the motion-blur video dataset obtained in Section 3.5. As this motion-blur video dataset addresses the impact of motion blur and does not consider other data augmentation operations, a linear kernel was chosen. Figure 12 shows the recorded error between the inter-frame velocity detected by this method and the set inter-frame velocity, as well as the overall displacement error.

By comparing the speed errors and displacement errors of different exposure times in Figure 12, it can be found that as the exposure time increases, there will be more significant speed errors and displacement errors during acceleration and deceleration. During the entire acceleration period, a maximum displacement error of −346 px was generated (when the exposure duration is 0.5 times that of the inter-frame time). During the uniform motion phase, no new displacement errors were generated, while during the deceleration phase, a total of 266 px maximum displacement errors were generated. After an acceleration uniform deceleration operation, the total error of the object shows the same trend as the detection results of FAST-PCC, but does not reach 0. The speed error and maximum displacement error during acceleration or deceleration are both related to the exposure time and have a nearly linear relationship. During acceleration, the speed error is negative, while during deceleration, the speed error is positive. However, the fluctuation of the velocity error and maximum displacement error detected by the improved KCF tracking algorithm exceeds that detected by FAST-PCC.

### 4.3. Experimental Analysis

Since visual tracking methods are based on the comparison of characteristic values (both the generation method and the discrimination method are the same, see Figure 1), it can be seen from the generation method of the PSF analyzed in Section 3.1 that if the length of motion blur is *L*, the motion-blur image only carries the image information of the position moved to at the exposure cut-off moment by $\frac{1}{2L}$, and it is difficult to track the actual position by this. If the motion is one-dimensional and the object motion velocity is assumed to remain constant during the exposure time, the expectation of the difference between the motion velocity detected by the correlation method and the actual velocity between the two frames should be calculated as follows:(62)$$E\left(v_{error}\left(i\right)\right)=\frac{L_{mb}\left(i\right)-L_{mb}\left(i-1\right)}{2}$$

For the accelerated motion whose velocity starts from 0, the expectation of the difference between the displacement of Frame i and the actual displacement detected by the correlation method is:(63)$$E\left(s_{error}\left(i\right)\right)=\sum_{j=1}^{i}E\left(v_{error}\left(i\right)\right)=\frac{L_{mb}\left(i\right)}{2}$$

This is consistent with the experimental results in Section 4.2, that is, the velocity error of a frame in MBHP is related to the motion-blur parameters of two consecutive frames.

**The cause of MBHP:** the motion-blur image carries too little information about the position of the exposure cut-off time, which cannot support the accurate tracking of the visual target tracking method.

**The main factors influencing MBHP:** the difference of the motion-blur length and the maximum motion-blur length of two consecutive frames.

**Consequences of the MBHP:** For certain and indeterminable templates (such as track by detecting), the accumulated errors on the displacement are detected, and the size of the accumulated errors is related to the motion-blur parameters before and after the motion. For detecting a semi-definite template (that is, a template that has timely updates), MBHP may contaminate the template, thus causing uncertainty of location.

## 5. Solutions for Visual Tracking with MBHP

According to the above analysis, MBHP will produce speed differences, so compensation values based on the speed error value can be added to the video motion detection results.

### 5.1. Estimation of Compensation Value of Speed Error Caused by MBHP

#### 5.1.1. Compensation Value Based on Inter-Frame Displacement Estimation Error

According to the approximate linear relationship between inter-frame speed and motion-blur length, the expected value of the compensation value can be calculated as follows:(64)$$E\left(v_{error}\left(i\right)\right)=\frac{L_{mb}\left(i\right)-L_{mb}\left(i-1\right)}{2}\approx \frac{T_{e}(v_{pf}\left(i\right)-v_{pf}\left(i-1\right))}{{2T}_{c}}$$
where $T_{e}$ is the exposure market of each frame, $T_{c}$ is the interval of each frame, and $v_{pf}$ is the interframe speed $v_{f}$ predicted by the video motion detection method.

#### 5.1.2. Compensation Value Based on PSF Sub-Pixel Estimation Method (SPEPSF)

The PSF sub-pixel estimation method based on cepstrum-based posteriori has been mentioned in Section 3.2. The motion-blur parameters of each frame are estimated by Formula (36), then the expected value of the compensation value can be calculated as follows:(65)$$E\left(v_{error}\left(i\right)\right)=\frac{L_{mb}\left(i\right)-L_{mb}\left(i-1\right)}{2}=E(\frac{L_{e}\left(i\right)-L_{e}\left(i-1\right)}{2})$$
where $L_{e}\left(i\right)$ is the motion-blur length of Frame i estimated by the SPEPSF method. Figure 13 shows the error of the motion-blur length of the motion-blur dataset proposed in Section 3.5, estimated by the SPEPSF method. The error is larger when the motion-blur length is small (<5 px), and the remaining maximum error is −1.1 px, with the maximum relative error rate of 0.2%.

#### 5.1.3. Compensation Value Based on No-Reference Image Quality Assessment Indicators

A uniformly accelerated image dataset with a motion-blur length ranging from 0 px to 600 px with an interval of 0.5 px is constructed from the same clear image in Section 3.5. The inter-frame speed is defined as:(66)$$v_{f}=n_{f}-1\;\;\;\;1\leq n_{f}\leq 1201$$
(67)$$L_{mb}=0.5v_{f}$$

The scores of the six NR-IQA indicators introduced in Section 2.3 are calculated, respectively, using this dataset and shown in Figure 14. It is worth noting that the average score of each pixel is used to adapt to pictures of different sizes. In addition, the pictures are first converted into gray images of floating point numbers with a gray value of 0–1, and then the gray value is amplified 100 times.

It can be clearly perceived from Figure 14 that for the dataset with a continuously increasing motion-blur length, scores of different NR-IQA indicators are different, and the score image of each quality assessment indicator is similar to y=1/x. In order to obtain general fitting results, the score of each indicator was multiplied by the power of the motion-blur length, making it linearly mapped to y=x, and the MSE of it and the straight line y=x was calculated for comparison as Formula (68).
(68)$${MSE}_{method}=min(\frac{1}{1201}\sum_{L=0,∆L=0.5}^{600}{\left(\frac{{Score}_{method}\times L^{z}}{c}-L\right)}^{2})$$

The parameters and MSE fitted by each method are shown in Table 4.

It can be seen from Table 4 that the sorting result from small to large based on MSE is: Laplacian, Tenengrad, SMD, EOG, Brenner, and SMD2. The MSEs of Laplacian, Tenengrad, and SMD are one or two orders of magnitude lower than the MSEs of the last three methods. So, Laplacian, Tenengrad, and SMD indicators are applied to the motion-blur length estimation of the dataset in Section 3.5 and the results are shown in Figure 15.

From Figure 15, it can be seen that the motion-blur length estimated by the Laplacian indicator is smaller than the error based on SMD and Tenengrad under three exposure durations, all within ±15 px. Based on this, it is recommended to use the Laplacian-based NR-IQA indicator to estimate and calculate the compensation value of MBHP. The estimated compensation value is calculated as follows:(69)$$E\left(v_{error}\left(i\right)\right)=\frac{L_{mb}\left(i\right)-L_{mb}\left(i-1\right)}{2}=\frac{L_{Le}\left(i\right)-L_{Le}\left(i-1\right)}{2}$$
where $L_{Le}\left(i\right)$ represents the motion-blur length of Frame i, estimated by using Laplacian scores fitting images with different motion-blur lengths.

### 5.2. Implementation and Comparison of Compensation Values

The paradigm of visual tracking detection (shown in Figure 1) is modified based on error compensation for MBHP, and the modified visual tracking flowchart is shown in Figure 16. Compared to Figure 1, Figure 16 adds an MBHP compensation value loop and the detection target is directly set to the change value of displacement.

The compensation values for the inter-frame velocity were calculated according to Figure 16, and Formulas (64), (65), and (69), and we compensated for the video tracking results calculated based on FAST-PCC in Section 4.2.1. The compensated velocity error and total displacement error are shown in Figure 17, Figure 18 and Figure 19, where Figure 17 represents the compensated result when the exposure time is 0.1 times that of the inter-frame time, Figure 18 represents the compensated result when the exposure time is 0.3 times that of the inter-frame time, and Figure 19 represents the compensated result when the exposure time is 0.5 times that of the inter-frame time. It can be observed that the speed error detected after adding speed compensation values tends to be 0, and the displacement error is significantly reduced compared to Figure 11. The maximum displacement error of the three compensation methods is only 9.5 px (the compensation value estimated based on the NR-IQA indicator is the result after compensation when the exposure time is 0.5 times that of the inter-frame time), which is 96.8% less than the maximum displacement error before compensation (300 px, Figure 11).

The three compensation values mentioned above are sufficient for their respective advantages and limitations. A compensation value based on an inter-frame displacement estimation error depends on the accuracy of displacement detection, and it is necessary to know the exposure time information in advance, which is not applicable to some incomplete image sequences. Compensation values based on SPEPSF can have good results for the original image or colored noise with Gaussian-white noise or noise with even no fixed frequency. However, for lossy compressed images, the residual frequency information after applying cepstrum transformation will interfere with the estimation of the PSF. Compensation values based on NR-IQA indicators will be affected by other fuzzy forms of interference. Therefore, for general motion-blur scenes, the compensation value of MBHP can be selected according to the specific situation.

## 6. Discussion and Conclusions

In this work, the motion-blur hysteresis phenomenon (MBHP) in visual tracking and displacement detection was discovered, which can cause cumulative errors and consequences in target tracking and the detection of objects with long-term acceleration or deceleration movements. By re-modeling the video shooting process, the APSF method was obtained. Using a cepstrum-based subpixel PSF estimation method (SPEPSF), it was verified that the MBO generated, based on an APSF method, has a smaller and more stable error compared to the actual MBO than the existing MBO generation methods. The overall error of MSE decreased by 49.47–77.82%, and the ME of the strong motion blur with a motion-blur length above 20 px decreased by 86–87%. Using the characteristics of long self-similarity objects and the APSF method, a motion-blur video dataset that is closer to the actual motion is constructed.

The APSF method may seem complex, but it has been packaged as MATLAB functions. Its source code has been added to the link in the Data Availability Statement. In addition, it can be interchanged with the built-in PSF generation function in MATLAB. Other developers are also welcome to develop it again.

MBHP was replicated in tracking experiments on a motion-blur video dataset using FAST-PCC and improved KCF tracking methods. The replication experiments showed that the error in the detected inter-frame velocity caused by MBHP is close to half of the difference of the motion-blur length between two consecutive frames. Therefore, it is particularly evident in motion with long time acceleration or deceleration and high absolute values of inter-frame acceleration.

Here, we explored the causes, impacts, and possible consequences of MBHP. The reason for MBHP is that the motion blurred image carries too little image information about the exposure cut-off position, which cannot support the accurate tracking of visual object tracking methods. The main factors affecting MBHP are the difference in the motion-blur length between two consecutive frames and the maximum motion-blur length. The consequence of MBHP is that for the tracking of fixed and uncertain templates (such as track by detection), it will cause cumulative errors in the displacement, and the magnitude of the cumulative errors is related to the motion-blur parameters before and after the motion. For the tracking of semi fixed templates (i.e., templates that are updated in a timely manner), MBHP will cause template contamination, resulting in positioning uncertainty. In reality, the impact of this phenomenon can be reduced by setting a smaller exposure time and reducing the relative speed between the camera and the object.

Finally, a general flow chart of visual tracking displacement detection with error compensation for MBHP is also included in this manuscript in Figure 16. It can be easily applied to other visual tracking methods and applications. MBHP suppression methods were proposed from three perspectives: compensation values based on inter-frame displacement estimation errors, SPEPSF, and NR-IQA indicators to weaken the impact of MBHP and facilitate selection in different tasks and environments. The experimental results show that the proposed MBHP error suppression method can reduce MBHP errors by more than 96%.

This work has a good complementary effect on detection tasks involving motion blur, especially those involving long-term acceleration and deceleration motions and strong motion blur. From a higher dimension, it can improve motion detection accuracy while maintaining a high image resolution, that is, it can improve the temporal resolution while maintaining a high spatial resolution. However, due to the differences in real scenarios, we did not find a permanent method to suppress MBHP, but instead used three different suppression methods. We hope to find a more robust solution in the future.

## Figures and Tables

**Figure 1 sensors-23-08024-f001:**
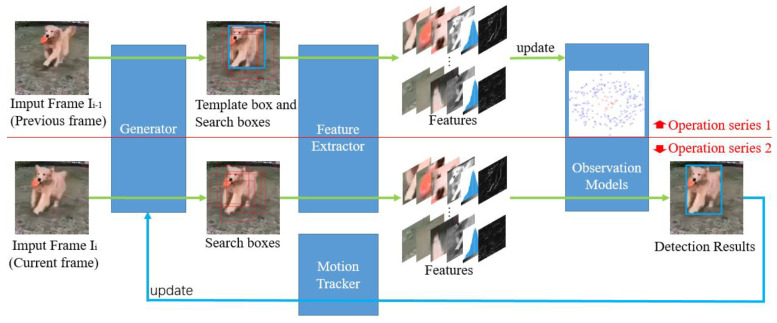
General flow chart of visual object tracking.

**Figure 2 sensors-23-08024-f002:**
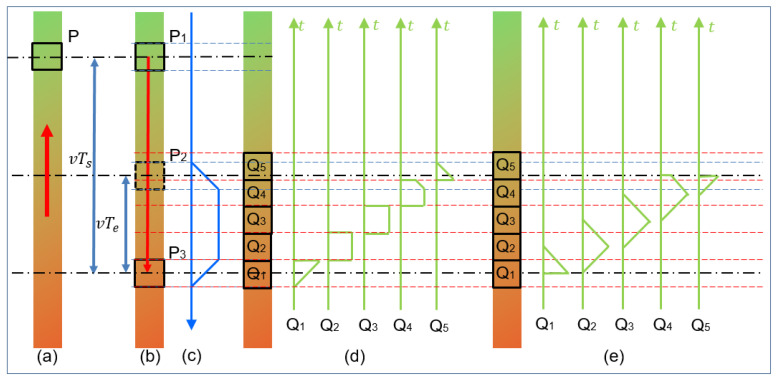
One-dimensional motion-blur process and decomposition diagram. (**a**) Observing real motion with pixel box P, (**b**) Observing pixel box with real object image, (**c**) Continuous one-dimensional motion-blur operator, (**d**) Ratio of one-dimensional motion-blur operators divided by region for each pixel point after discretization of real object image, (**e**) Ratio of one-dimensional motion-blur operators divided by time for each pixel point after discretization of real object image.

**Figure 3 sensors-23-08024-f003:**
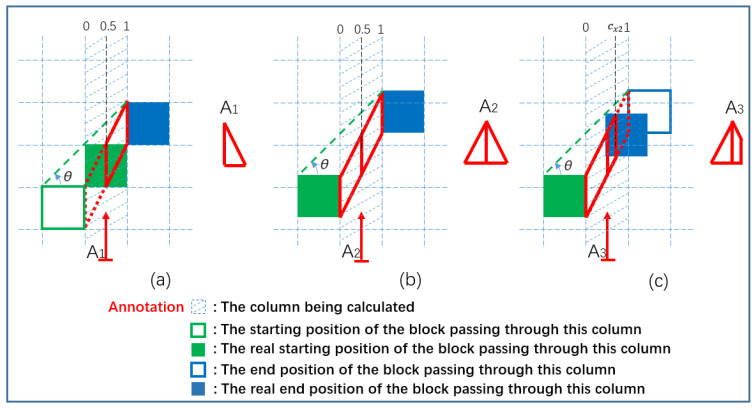
Schematic diagram of a column decomposition of a two-dimensional MBO. (**a**) The real starting position that passes through this column is in this column, and the ending position is in the right column. (**b**) The starting position that passes through this column is in the left column, and the ending position is in the right column. (**c**) The starting position of the blocks that pass through this column is in the left column, and the ending position is between the left and right columns.

**Figure 4 sensors-23-08024-f004:**
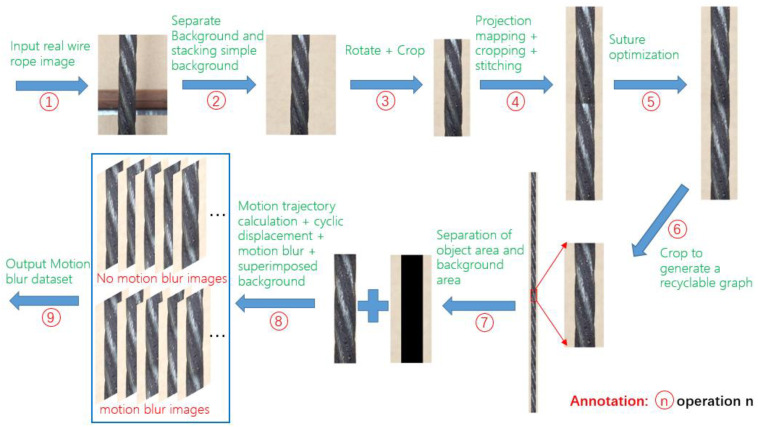
Flow chart of motion-blur video dataset generation based on MBO generation method and self-similarity object images.

**Figure 5 sensors-23-08024-f005:**
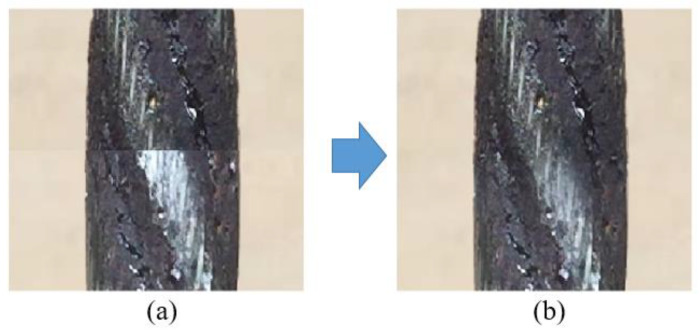
Comparison diagram of linear transition treatment before and after splicing. (**a**) Before processing, (**b**) After processing.

**Figure 6 sensors-23-08024-f006:**
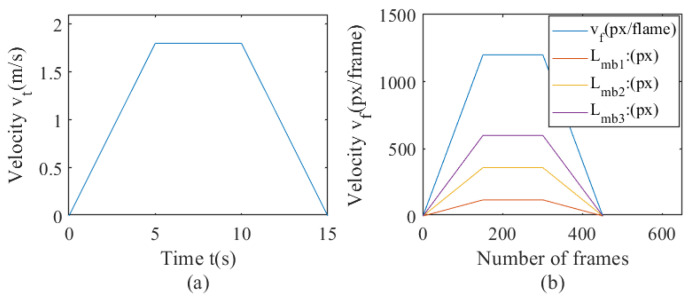
Graph of setting speed and motion blur length; (**a**) the graph of speed and time, (**b**) the graph of speed and frame number, and the graph of motion blur length and frame number.

**Figure 7 sensors-23-08024-f007:**
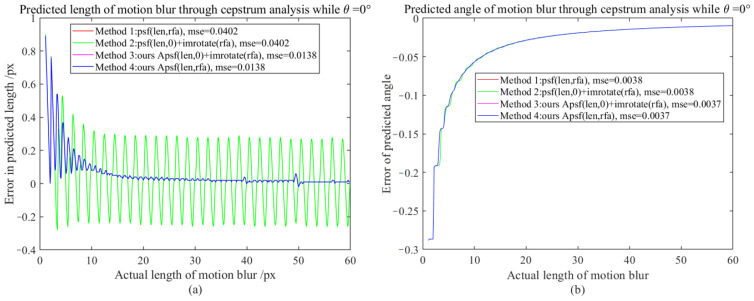
The motion-blur length and angle error map when the motion-blur angle is 0°; (**a**) Motion-blur length error graph, (**b**) Motion-blur angle error graph.

**Figure 8 sensors-23-08024-f008:**
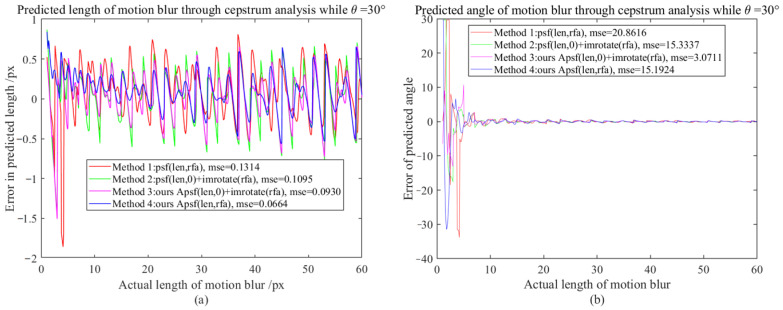
The motion-blur length and angle error map when the motion-blur angle is 30°; (**a**) Motion-blur length error graph, (**b**) Motion-blur angle error graph.

**Figure 9 sensors-23-08024-f009:**
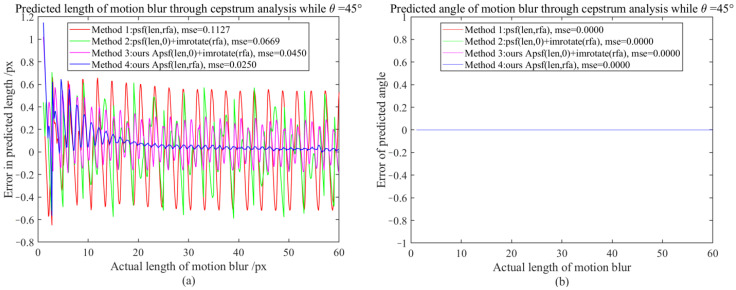
The motion-blur length and angle error map when the motion-blur angle is 45°; (**a**) Motion-blur length error graph, (**b**) Motion-blur angle error graph.

**Figure 10 sensors-23-08024-f010:**
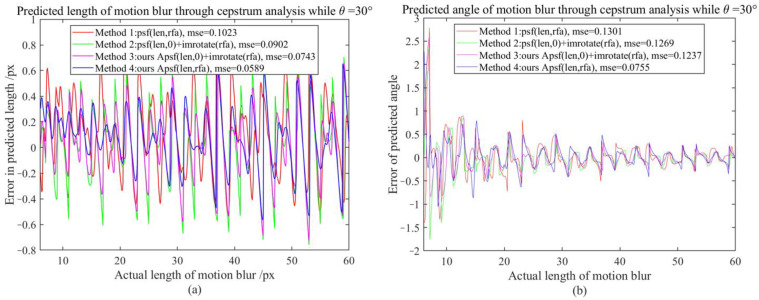
The motion-blur length and angle error map when the motion-blur angle is 30° with the motion-blur length range from 6–60 px; (**a**) Motion-blur length error graph, (**b**) Motion-blur angle error graph.

**Figure 11 sensors-23-08024-f011:**
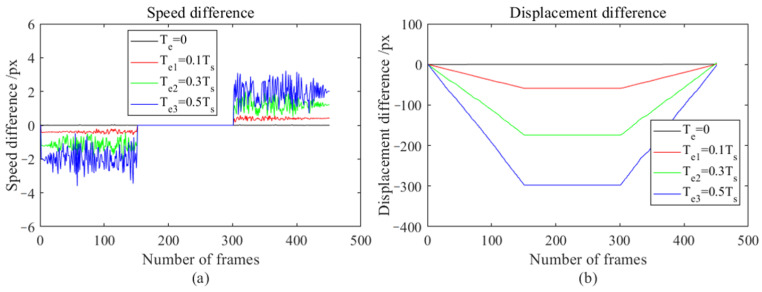
Error plots of speed and displacement detected by FAST-PCC under different exposure times; (**a**) Speed error chart, (**b**) Displacement error chart.

**Figure 12 sensors-23-08024-f012:**
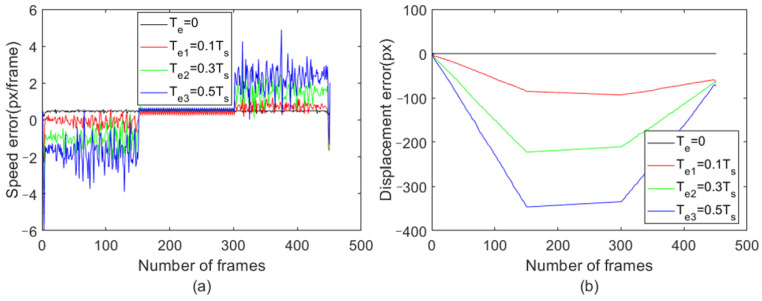
Error plots of speed and displacement detected by improved KCF under different exposure times; (**a**) Speed error chart, (**b**) Displacement error chart.

**Figure 13 sensors-23-08024-f013:**
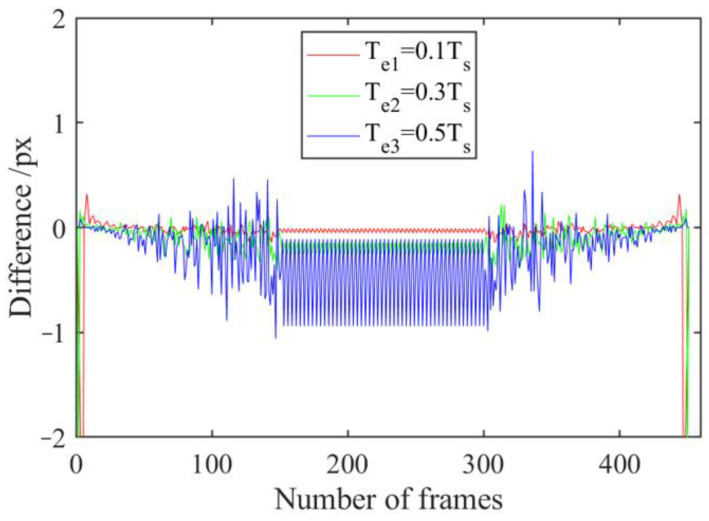
Error plot of motion-blur length of motion-blur dataset based on SPEPSF.

**Figure 14 sensors-23-08024-f014:**
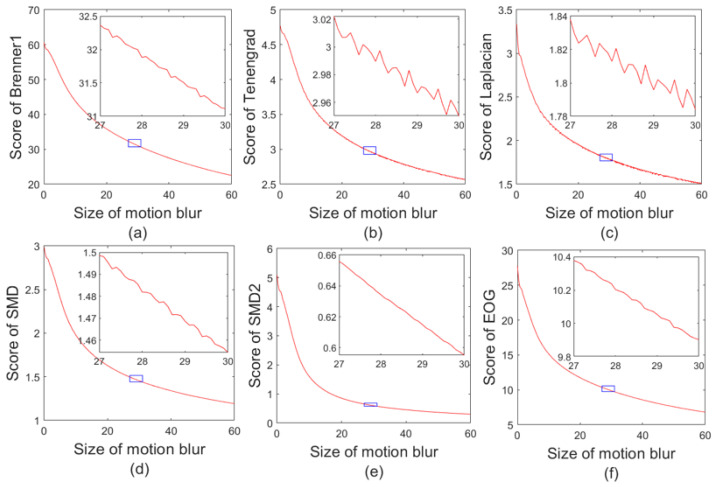
Score graphs for six NR-IQA indicators. (**a**) Brenner, (**b**) Tenengrad, (**c**) Laplacian, (**d**) SMD, (**e**) SMD2, (**f**) EOG.

**Figure 15 sensors-23-08024-f015:**
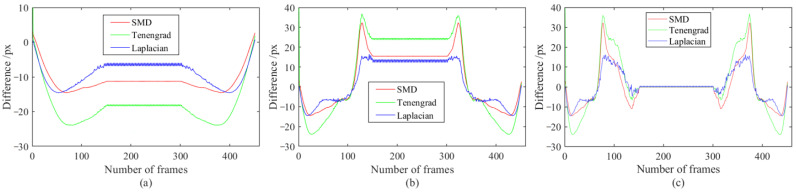
Error plot of motion-blur length estimated based on NR-IQA indicators under different exposure times. The exposure time is (**a**) 0.1, (**b**) 0.3, and (**c**) 0.5 times that of the inter-frame time.

**Figure 16 sensors-23-08024-f016:**
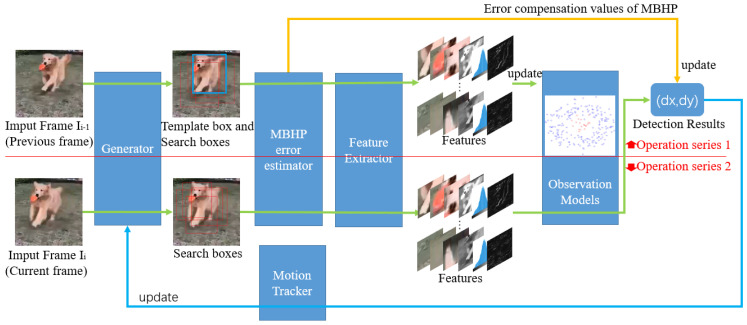
General flow chart of visual tracking displacement detection with error compensation for MBHP.

**Figure 17 sensors-23-08024-f017:**
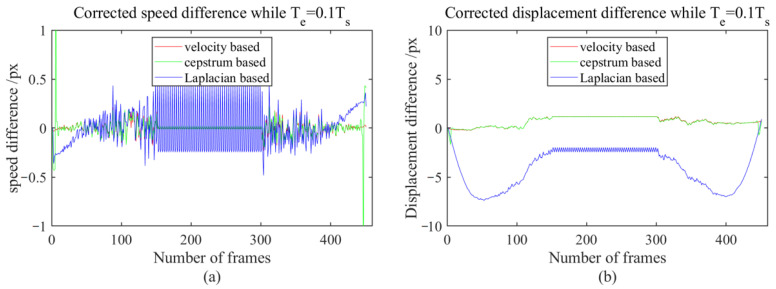
Compensated velocity error and total displacement error plot when exposure time is 0.1 times that of the inter-frame time; (**a**) velocity error plot, (**b**) displacement error plot.

**Figure 18 sensors-23-08024-f018:**
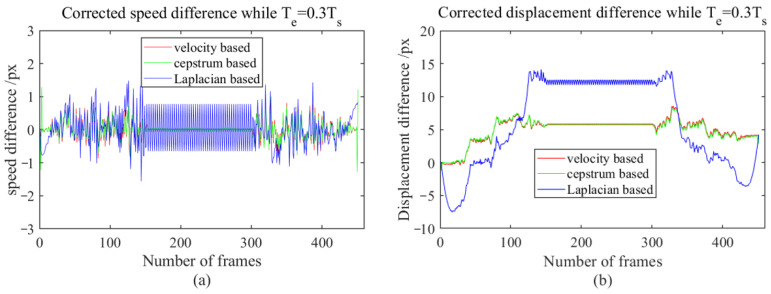
Compensated velocity error and total displacement error plot when exposure time is 0.3 times that of the inter-frame time; (**a**) velocity error plot, (**b**) displacement error plot.

**Figure 19 sensors-23-08024-f019:**
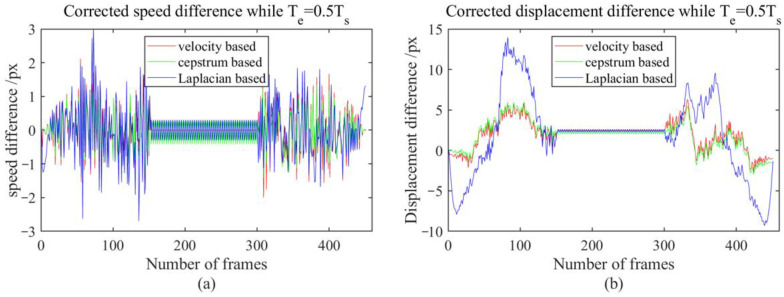
Compensated velocity error and total displacement error plot when exposure time is 0.5 times that of the inter-frame time; (**a**) velocity error plot, (**b**) displacement error plot.

**Table 1 sensors-23-08024-t001:** The table of the MBO generated by Formula (8) with L=6, θ=45.

	1	2	3	4	5
1	0	0	0	0.044	0.100
2	0	0	0.044	0.150	0.044
3	0	0.044	0.150	0.044	0
4	0.044	0.150	0.044	0	0
5	0.100	0.044	0	0	0

**Table 2 sensors-23-08024-t002:** The table of the MBO generated by APSF with L=6, θ=45.

	1	2	3	4	5	6
1	0	0	0	0	0.039	0.079
2	0	0	0	0.039	0.157	0.039
3	0	0	0.039	0.157	0.039	0
4	0	0.039	0.157	0.039	0	0
5	0.006	0.123	0.039	0	0	0
6	0.001	0.006	0	0	0	0

**Table 3 sensors-23-08024-t003:** The comparison of four MBO generation methods in terms of ME and MSE.

Method	θ=0		θ=30		θ=45	
	ME(L≥20)	MSE	ME(L≥20)	MSE	ME(L≥20)	MSE
Method 1	0.29	0.0402	0.80	0.1314	0.60	0.1127
Method 2	0.29	0.0402	0.75	0.1095	0.58	0.0669
Method 3	0.04	0.0138	0.72	0.0930	0.32	0.0450
Method 4	0.04	0.0138	0.64	0.0664	0.08	0.0250

**Table 4 sensors-23-08024-t004:** The parameters and MSE fitted by each NR-IQA indicator.

Indicator	Brenner	Tenengrad	Laplacian	SMD	SMD2	EOG
z	1.704	1.251	1.235	1.383	2.442	1.704
c	177.5764	12.1532	7.1211	10.8109	217.9918	177.5764
MSE	119.1523	21.4488	5.0214	22.2483	827.8391	119.1523

## Data Availability

The motion-blur video dataset and the source code of the APSF method can be found here: (https://pan.baidu.com/s/1TEYohk2sXRnjqZqLTCmA6g?pwd=MBHP, accessed on 1 September 2023).

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
