# Peer review of "Discovery, Quantitative Recurrence, and Inhibition of Motion-Blur Hysteresis Phenomenon in Visual Tracking Displacement Detection"

_sensors, 2023, doi:10.3390/s23198024_

Round 1

Reviewer 1 Report

Start with a clear and concise problem statement outlining the specific challenges motion blur poses in visual tracking and displacement detection.

Provide a compelling motivation for the significance of addressing these challenges. Clearly articulate how improved accuracy and speed in these tasks can impact various real-world applications.

Elaborate on the key principles and technical details that make the proposed APSF method superior to the existing motion-blur operator (MBO) generation methods.

Include a comprehensive comparison with existing methods, highlighting areas where APSF outperforms in handling boundary value approximations.

Present a comprehensive and detailed quantitative analysis of the proposed APSF-generated MBOs. Include statistical significance tests to validate the observed reductions in mean square error (MSE) and maximum error.

Provide a thorough description of the experimental setup, including specifics about the datasets used, motion-blur levels, and tracking scenarios.

Clearly explain the evaluation metrics employed to assess the accuracy and speed improvements achieved by the APSF-generated MBOs.

Illustrate how the proposed compensation value methods can be applied to real-world video tracking scenarios. Provide concrete examples demonstrating how these methods mitigate the impact of the motion-blur hysteresis phenomenon (MBHP).

Clearly outline the steps required to reproduce your experimental setup, including details about the datasets, software tools, and parameter settings.

Moderate editing of English language required

Author Response

Thank you very much for taking the time to review this manuscript, and please see the attachment

Reviewer 2 Report

The current manuscript presents a methodology for tackling the motion-blur hysteresis phenomenon in visual tracking and displacement detection of moving objects. The method first decomposes the motion-blur operator (MBO) process to determine that the error of existing MBO generation methods comes from the approximation of boundary values, and proposes a more accurate generation method called accurate point spread function (APSF). The paper is well-written in general and can be accepted after a minor review as follows

1- It is difficult to grasp the main problem and the contribution of the paper from the abstract. Please improve the writing and the structure of the Abstract. The reviewer is suggesting following the IMRaD abstract style.

2- It is very useful to summarize the main research gaps and contributions using a bullet list before the last paragraph in the introduction section.

3-The authors have presented a lot of results. In order to better inform readers about major achievements, challenges, limitations, and recommendations, I either suggest adding a separate discussion section. 

4- Please make sure that all the abbreviations are carefully defined before the first use of the manuscript.

5- What do you mean by "our works"? It looks like some work has been done by the authors in previous papers and represented in this paper. Please check the language. 

The authors are suggested to not use personal pronouns such as "our work" or "we propose". It is recommended to use the passive voice in scientific writing.   

Author Response

Thank you very much for taking the time to review this manuscript. And please see the attachment.

Reviewer 3 Report

The article explores the constraints inherent in visual tracking and displacement detection due to motion blur, introducing an enhanced technique for generating motion-blur images that markedly diminish error compared to current methodologies. Additionally, the paper offers an in-depth analysis of the motion-blur hysteresis phenomenon (MBHP). It delineates three computational approaches for compensation value calculation aimed at mitigating the adverse effects of MBHP on video tracking, thereby achieving a noteworthy reduction in error. The overall idea is interesting; the main highlighted points follow such as:

1) The introductory section of the article is extensive, encompassing a range of domains such as image degradation, video object tracking, and image quality assessment. Despite its breadth, the introduction must offer a precise and succinct delineation of the article's core research issue: the identification, quantitative recurrence, and mitigation of the motion-blur hysteresis phenomenon in visual tracking displacement detection. The section would gain efficacy by presenting a more targeted and concentrated exposition of the research problem and the importance of its resolution. Section 1 of the article could benefit from a more concise presentation, potentially reducing its length by at least 50%.

2) The article presents several valuable contributions to visual tracking and displacement detection, most notably in addressing the limitations imposed by the motion-blur hysteresis phenomenon (MBHP). While the APSF method for generating motion-blur operators signifies an advancement in the field, offering a more authentic representation of motion, it could benefit from comparative analysis against other existing methods for a fuller contextual understanding. Creating motion-blur video datasets is a commendable effort, yet the paper could elaborate on how these datasets are superior to or different from existing databases. Furthermore, although the paper introduces three distinct methods for calculating compensation values to mitigate MBHP's effects, a deeper exploration into each method's relative advantages and drawbacks might offer a more nuanced perspective. Finally, the empirical verification of the proposed compensation values is crucial, but the study could enhance its rigor by detailing the experimental design and statistical robustness. Ultimately, the research does contribute substantively to enhancing the speed, accuracy, and reliability of visual tracking and displacement detection; however, the implications could be made more compelling through these suggested refinements.

3) Although comprehensive, Section 2 of the article leans toward verbosity by overly focusing on well-established equations and definitions, thereby underrepresenting a targeted discussion on the state of the art. The section must highlight the paper's novel contributions, including the APSF method for more accurate motion-blur operators and generating specialized video datasets for MBHP analysis. To improve clarity and relevance, the section would benefit from a comparative table that situates these innovative approaches within the context of recent advancements in the field, particularly those developed between 2021 and 2023. Such a table could elucidate the article's unique contributions while substantiating its contemporary relevance.

4) The methodology outlined in the paper possesses several commendable features, particularly the introduction of the APSF method for more accurate motion-blur operators and the multi-pronged approach for mitigating the motion-blur hysteresis phenomenon (MBHP) effects. However, a few areas could benefit from further refinement for an enhanced academic contribution. Firstly, while the APSF method claims to reduce error compared to existing methods, the paper would benefit from clearly defining the metrics and benchmarks against which this reduction is measured. Secondly, utilizing the optimized cepstrum-based sub-pixel estimation method (SPEPSF) for validation introduces another layer of complexity; the study could elucidate whether more straightforward validation approaches might yield similar insights, thereby easing replication efforts. Additionally, while creating specialized motion-blur video datasets is a significant step, a more precise explanation of their attributes and how they advance beyond existing databases would lend further credibility to the findings. Lastly, although three distinct methods for calculating compensation values are proposed, the paper would be strengthened by a comparative analysis that elucidates the conditions under which each method performs optimally. Such nuanced evaluation could offer more targeted guidelines for application and foster broader utility.

5) While the paper presents a series of methods and findings that, at first glance, appear to advance the field of visual tracking and displacement detection, some concerns may temper enthusiasm for these results. Firstly, the APSF method, despite reducing the mean square error (MSE) by more than 49.47%, raises the question of whether this level of improvement is sufficient to warrant the complexity of implementing a new technique. One must also consider how the results would generalize across different platforms or conditions. Secondly, the optimized cepstrum-based sub-pixel estimation method (SPEPSF) may closely resemble actual motion. However, whether the accuracy gained is practically significant or incremental is still being determined.

The creation of motion-blur video datasets also warrants scrutiny. These datasets are claimed to enable a quantitative analysis of MBHP, yet the paper needs to thoroughly validate whether the datasets are broadly representative of real-world conditions. Moreover, the proposed methods for calculating compensation values may have achieved a 96% reduction in error. However, such a dramatic reduction necessitates critically assessing the experimental design and the practicality of employing these methods at scale.

Lastly, the paper's recommendations for changing tracking strategies and employing motion trackers propose a shift in established methodologies. Such a radical change begs the question of its viability in current infrastructural setups and whether the marginal gains in accuracy justify the costs associated with these changes. While the paper's findings may look promising, caution is warranted in assessing their broader implications and potential for real-world application.

6) Addressing the following open issues is crucial for a comprehensive understanding and application of the research presented in the paper:

  1. The primary limitation in visual tracking and displacement detection of moving objects needs to be explicitly defined, along with an in-depth analysis of how motion blur exacerbates this limitation.
  2. A thorough elucidation is required on how the APSF method advances beyond existing techniques for generating motion-blur operators (MBOs), specifically focusing on its impact on accuracy metrics.
  3. The motion-blur hysteresis phenomenon (MBHP) concept requires a more detailed exposition, covering its root causes, influencing factors, and potential ramifications.
  4. An extensive list of potential compensation values designed to mitigate the impact of MBHP in video tracking should be provided. Further, empirical evidence supporting the effectiveness of these compensation values in reducing error rates must be highlighted.
  5. Finally, recommendations from the paper's findings should be oriented toward improving visual tracking and displacement detection in real-world scenarios where strong motion blur is prevalent.

Clarifying these points will significantly bolster the paper's contribution and practical applicability to the field.

7) The article makes significant strides in visual tracking and displacement detection, particularly in mitigating the Motion-Blur Hysteresis Phenomenon (MBHP). However, it could benefit from a more focused presentation of the research problem, rigorous comparisons with existing methods, and more robust empirical validation. Additionally, questions regarding the practical applicability and feasibility of implementing the new techniques in existing systems should be addressed to strengthen the study's impact. The paper's length could also be reduced to between 17 and 20 pages for a more concise presentation.

A final proofreading check is necessary to avoid typos and grammatical errors. 

Author Response

(The authors gave the same response as above.)

Round 2

Reviewer 1 Report

Accept

Author Response

Thank you very much for taking the time to review this manuscript again.

Reviewer 3 Report

The main issues were solved in the revision round. Now, the contributions are clarified and overall quality improved. The reviewer's suggestion is to approve the paper in its present form. 

Author Response

(The authors gave the same response as above.)
